# Combined Effects of HA Concentration and Unit Cell Geometry on the Biomechanical Behavior of PCL/HA Scaffold for Tissue Engineering Applications Produced by LPBF

**DOI:** 10.3390/ma16144950

**Published:** 2023-07-11

**Authors:** Maria Laura Gatto, Michele Furlani, Alessandra Giuliani, Marcello Cabibbo, Nora Bloise, Lorenzo Fassina, Marlena Petruczuk, Livia Visai, Paolo Mengucci

**Affiliations:** 1Department of Industrial Engineering and Mathematical Sciences, Polytechnic University of Marche, Via Brecce Bianche 12, 60131 Ancona, Italy; m.l.gatto@univpm.it (M.L.G.); m.cabibbo@univpm.it (M.C.); 2Department of Clinical Science, Polytechnic University of Marche, Via Brecce Bianche 12, 60131 Ancona, Italy; a.giuliani@univpm.it; 3Department of Molecular Medicine, Centre for Health Technologies (CHT), INSTM UdR of Pavia, University of Pavia, Viale Taramelli 3/b, 27100 Pavia, Italy; nora.bloise@unipv.it (N.B.); livia.visai@unipv.it (L.V.); 4Medicina Clinica-Specialistica, UOR5 Laboratorio di Nanotecnologie, ICS Maugeri, IRCCS, Via Salvatore Maugeri 4, 27100 Pavia, Italy; 5Department of Electrical, Computer and Biomedical Engineering, Centre for Health Technologies (CHT), University of Pavia, Via Ferrata 5, 27100 Pavia, Italy; lorenzo.fassina@unipv.it; 6Faculty of Materials Science and Engineering, Warsaw University of Technology, 141 Woloska Str., 02-507 Warsaw, Poland; marlena.petruczuk.stud@pw.edu.pl; 7Department of Materials, Environmental Sciences and Urban Planning, INSTM UdR of Ancona, Polytechnic University of Marche, Via Brecce Bianche 12, 60131 Ancona, Italy; p.mengucci@univpm.it

**Keywords:** polycaprolactone/hydroxyapatite scaffold, hydroxyapatite concentration, unit cell geometry, laser powder bed fusion, tissue engineering, Haralick texture analysis, mechanical performances, human mesenchymal stem cells

## Abstract

This experimental study aims at filling the gap in the literature concerning the combined effects of hydroxyapatite (HA) concentration and elementary unit cell geometry on the biomechanical performances of additively manufactured polycaprolactone/hydroxyapatite (PCL/HA) scaffolds for tissue engineering applications. Scaffolds produced by laser powder bed fusion (LPBF) with diamond (DO) and rhombic dodecahedron (RD) elementary unit cells and HA concentrations of 5, 30 and 50 wt.% were subjected to structural, mechanical and biological characterization to investigate the biomechanical and degradative behavior from the perspective of bone tissue regeneration. Haralick’s features describing surface pattern, correlation between micro- and macro-structural properties and human mesenchymal stem cell (hMSC) viability and proliferation have been considered. Experimental results showed that HA has negative influence on scaffold compaction under compression, while on the contrary it has a positive effect on hMSC adhesion. The unit cell geometry influences the mechanical response in the plastic regime and also has an effect on the cell proliferation. Finally, both HA concentration and elementary unit cell geometry affect the scaffold elastic deformation behavior as well as the amount of micro-porosity which, in turn, influences the scaffold degradation rate.

## 1. Introduction

The employment of scaffold as a cell transplant device for regenerating damaged tissue is a novel approach to bone tissue engineering. The scaffold, designed to guide cell organization and growth and to match the mechanical properties of the injury site, is expected to stimulate healing and ensure the full recovery of tissue functionalities [1]. Scaffold geometry plays a pivotal role in tailoring mechanical performance and biological response. A periodical arrangement of elementary unit cells in the 3D space (lattice) allows for the customization of the porous network, improving scaffold permeability and degradability and, at the same time, minimizing risk of stress shielding [2,3].

By the appropriate choice of biomaterial and scaffold design, the native extracellular matrix (ECM) functions can be reproduced, thus providing structural support and promoting a good healing response of host tissue [4].

Natural polymers possess a high degree of similarity to the ECM [5], although they are difficult to process into the required shapes while maintaining unaltered biological functions. In addition, physical and chemical variations peculiar to each production batch hinder mass production for biomedical use [6,7]. Therefore, synthetic polymers represent an attractive solution because of their physical–chemical and mechanical properties. In particular, polycaprolactone (PCL) has been widely used in many tissue engineering (TE) applications, as it is chemically inert, biocompatible, low-cost, bioabsorbable and FDA-approved. However, PCL shows few limitations in vivo, including hydrophobicity, which prevents cell attachment and proliferation, and slow degradation rate. To overcome such drawbacks, addition of calcium phosphate-based ceramics in PCL has led to a composite biomaterial with improved mechanical properties, controllable degradation rates and enhanced bioactivity [8,9,10,11]. Hydroxyapatite (HA) chemistry is similar to apatite in natural bone. In biological media, HA reacts with ions of the body fluid, forming a surface apatite coating which induces protein adsorption and cell attachment, stimulating bone formation and biomaterial resorption [12]. Therefore, the appropriate selection of materials and their combination help to improve the biomechanical response of the device to support bone regeneration.

Another way to enhance the biomechanical properties of composite scaffolds is the correct choice of the production process. Manufacturing methods should also be effective to minimize toxic residues and cost, thus allowing large-scale production of patient-customized bioresorbable devices [2]. Additive manufacturing (AM) technologies allow the precise design of the internal structure of the biomaterial according to the mechanical and biological requirements for tissue engineering. In the past decade, AM technologies have boosted the fabrication of customized PCL products, with shorter processing time and reduced material waste [13,14].

To the best of the authors’ knowledge, previous studies on the effectiveness of combining AM technologies with the biomechanical performances of PCL/HA scaffolds for bone tissue regeneration have separately focused on the effects of the geometry [15,16,17,18,19,20] or of the HA concentration [21,22,23].

Geometries already studied in the literature range from very simple structures [15,16,17,18,19] to complex geometries based on a unit cell topology which is regularly repeated in space [20]. Experimental results have indicated that geometries resulting from diamond (DO) and rhombic dodecahedron (RD) elementary unit cells are suitable for human bone regeneration in the trabecular mandibular region, with regard to bioresorbability and biomechanical response [20].

Although recent studies have focused on the impact of HA concentration on the biomechanical properties of PCL/HA scaffolds, the HA amount considered in such studies is limited to 30 wt.% or less [21,22,23]. Rezania et al. [21] fabricated PCL/HA scaffolds with 5, 10, 15 and 20 wt.% of HA, by the fused filament fabrication (FFF) technique, aiming to produce affordable medical tools. The range of HA percentages was chosen because of the rheology and printability of the filaments. Young’s modulus of PCL/HA 20 wt.% printed scaffold increased by 50% compared to PCL and all the samples had qualified cytocompatibility with a human osteoblast cell line without statistically significant differences. Liu et al. [22] produced PCL/HA scaffolds by the melting deposition forming method using PCL/HA composites with 5, 10, 15, 20 and 25 wt.% HA content. In their study, PCL/HA scaffold with 25 wt.% HA achieved the best comprehensive performance for biomedical applications. Kim et al. [23] fabricated PCL/HA composite filaments with ceramic particles at 5, 10, 15, 20 and 25 wt.%, to print scaffolds with an FDM-type 3D printer for bone regeneration purposes.

This experimental study aims to fill the gap in the literature, by focusing attention on the combined effect of geometry and HA amount on the biomechanical and degradative behavior of PCL/HA scaffolds produced by AM. Scaffolds with DO and RD elementary unit cell geometry, charged with HA amounts of 5, 30 and 50 wt.% and produced by the laser powder bed fusion (LPBF) technique, were submitted to structural, mechanical and biological characterization. Results clearly showed the impact of HA amount and unit cell geometry on the mechanical response as well as on the biological and degradative behavior.

## 2. Materials and Methods

### 2.1. Design and Production

Two different elementary unit cells with sides of 1 mm (Figure 1) were used, based on findings of Gatto et al. [20], to create diamond (DO) and rhombic dodecahedron (RD) scaffolds, filling a volume of 10 × 10 × 5 mm^3^, Figure 1. DO and RD scaffolds were designed (Materialise Magics software, vers. 21.0, Materialise, Belgium) with 80% nominal porosity and strut thickness of 0.42 mm and 0.46 mm, respectively.

Three different powder mixtures obtained by combining polycaprolactone (PCL, Eurocoating S.p.a., Pergine Valsugana, Italy) and hydroxyapatite (HA, Boc Sciences, Inc., Shirley, NY, USA) in 5, 30 and 50 wt.% were used for scaffold manufacturing by laser powder bed fusion (LPBF) technology, using a Formiga P110 Velocis (EOS GmbH, Munich, Germany) manufacturing system. Printing parameters were optimized, starting from findings of Gatto et al. [20].

From this point on, samples are indicated by geometry (DO and RD) and percentage of HA in wt.% (5, 30 or 50). As an example, the PCL-based scaffold produced with diamond geometry and 5 wt.% HA compound is referred to as DO5. The same wording is also applied to the powders (P).

### 2.2. Morphological and Structural Characterization

The morphology of PCL/HA powders, the scaffold surface and cells after 4 days of incubation on scaffolds were investigated by a Tescan Vega 3 (Tescan Company, Brno, Czech Republic) scanning electron microscope (SEM) equipped with an EDAX Elements energy dispersive micro-analysis (EDS) system. Based on SEM images of P50 powders, the normal size distribution of HA and PCL particles was calculated using FIJI software (Version 2.3.0) [24]. The chemical composition of powders and scaffolds was determined by EDS micro-analysis. Results were obtained by averaging data taken from five different sample areas observed at the same magnification (500×). Chemical concentration of calcium (Ca), phosphorus (P) and the Ca/P ratio were considered. Structural information from powders and scaffolds was achieved using X-ray diffraction (XRD), by using a Bruker D8 Advance diffractometer (Bruker, Karlsruhe, Germany) operating at V = 40 kV and I = 40 mA, with Cu-Kα radiation, in the angular range 2θ = 10°–60°. Pattern analysis was performed by the DIFFRAC.EVA software package (Version 4.3.0.1) (Bruker), while XRD peak shape analysis was conducted by the OriginPro 2023 software (OriginLab, Northampton, MA, USA).

### 2.3. Haralick’s Surface Analysis

Surface texture of scaffolds was studied through two Haralick’s surface features (energy and variance) [25], calculated starting from the SEM images obtained via secondary electrons (SEs) at 350× of magnification. This approach evaluates the surface texture using a gray-level image of the surface [26,27]. Briefly, for each SEM image of the scaffold surface, at least two regions of interest (ROIs) were selected to measure the gray-level co-occurrence matrix (GLCM) and then, for each GLCM, Haralick’s energy and variance values were quantified. The energy computes the amount of similarity inside the GLCM and is a measure of local homogeneity in pixel values, whereas the variance has the same meaning as the statistical variance and is a measure of the local heterogeneity in pixel values.

### 2.4. Roughness

The surface roughness of scaffolds was measured using a Nikon LV 150 Confovis microscope (Nikon, Tokyo, Japan). According to ISO 25178–603 [28], quantitative information was extracted from roughness maps. Surface roughness parameters under consideration were surface skewness (Ssk) and surface kurtosis (Sku).

### 2.5. XµCT

To quantify morphometric parameters, scaffolds were subjected to X-ray micro-computed tomography (XCT) analysis using a Bruker Skyscan 1174 tomographic system (Bruker, Billerica, MA, USA). As shown in Table 1, scaffold projections were obtained at V = 50 kV and I = 800 A using the experimental settings listed in Table 1. The conditions under which projections were processed in stacks of cross-sectional slices using Nrecon reconstruction software (version 1.7.1.6) (Bruker) are also detailed in Table 1.

DO and RD scaffolds for each compound composition were analyzed with CT-analyzer software (Version 1.18) (Bruker). Morphometric parameters examined were micro-porosity (%) and total porosity (%).

### 2.6. Mechanical Tests

Four samples of each scaffold typology were subjected to compressive testing employing an Instron 5567 machine (Instron, Norwood, MA, USA) with a 5 kN maximum load and 0.5 mm/min compression rate. Tests were stopped at 50% displacement from initial scaffold height. Results were plotted as applied load (N) vs. compression on initial sample height (%). In addition, compressive modulus © and ultimate compressive strength (σ_UC_) were quantified from the stress/strain curve, according to ASTM D 1621e10 [29].

### 2.7. Biological Tests

#### 2.7.1. Cell Culture and Seeding

Human bone marrow mesenchymal stem cells (hBM-MSCs) were isolated and phenotypically analyzed in accordance with the International Society for Cellular Therapy [30,31], to evaluate mesenchymal properties. The Institutional Review Boards of the Fondazione IRCCS Policlinico San Matteo and the University of Pavia (2011) authorized the study protocols. All participants provided their written, informed consent. In all the experiments, cells were mainly at passages 4–5. hBM-MSCs were cultured in a low-glucose Dulbecco’s modified Eagle’s medium (DMEM) as maintenance medium supplemented with 10% MesenCult, 2% glutamine, 1% penicillin–streptomycin (P-S) and 1% amphotericin B (Lonza Group Ltd., Basel, Switzerland) and maintained in an incubator at 37 °C with a 5% CO_2_ atmosphere. Before cell seeding, scaffolds were sterilized in a 70% ethanol bath for 20 min before being extensively rinsed with sterilized water and phosphate-buffered saline (PBS) solution. After 40 min of UV exposure, dried scaffolds were deposited in a 24-well ultralow cell attachment plate (Corning, Inc., Corning, NY, USA) and incubated overnight in the culture medium. Cell seeding density was 5 × 10^4^ cells/sample.

#### 2.7.2. Cell Viability

In order to assess cell viability, cell mitochondrial activity was evaluated after 24 h and 4 days of culture with a 3-(4,5-dimethylthiazole-2-yl)-2,5-diphenyl tetrazolium bromide assay (MTT; Sigma-Aldrich, St. Louis, MO, USA), as previously described [32]. Absorbance was measured at 570 nm using 100 μL samples and a CLARIOstar^®^ Plus Multi-mode Microplate Reader (BMG Labtech, Ortenberg, Germany). Interpolation of a titration curve was used to determine the number of cells in each sample. The results were reported as the mean ± standard deviation. Each biological experiment was conducted in triplicate and in a minimum of three distinct assays. Statistical analysis was carried out using GraphPad Prism 6.0 (GraphPad, Inc., San Diego, CA, USA). Analysis was performed using one-way or two-way analysis of variance (ANOVA), followed by a Bonferroni post hoc test (significance level of 0.05).

#### 2.7.3. Cell Morphology

After 4 days of incubation, hMSC-seeded scaffolds were processed to analyze cell morphology by SEM. Samples were fixed with a 2.5% (*v*/*v*) glutaraldehyde solution in a 0.1 M Na-cacodylate buffer (pH = 7.2) for 1 h at 4 °C, rinsed with Na-cacodylate buffer and then dehydrated at room temperature in an ethanol gradient series up to 100%. After 6 h of lyophilization for complete dehydration, samples were sputter-coated with gold for SEM observation.

### 2.8. PhC-XµCT

Interaction between cells and scaffold material after 24 h and 4 days from cell seeding was investigated with phase-contrast X-ray micro-tomography (PhC-XµCT). PhC-XµCT analysis was conducted at the SYRMEP beamline of the ELETTRA Synchrotron Radiation Facility (Trieste, Italy).

While conventional XµCT analysis presents results based on attenuation contrast, PhC-XµCT imaging relies on phase contrast, according to the following Equation (1):n = −1 − δ + iβ (1)
where n is the refractive index of material. Since δ, the real decrement of the refractive index n, is one hundred times higher than the imaginary part β in soft tissues, the phase contrast approach provides higher sensitivity than attenuation contrast for the investigated samples.

The experimental setup, fine-tuned to better discriminate PCL, HA and background, was: white X-ray beam with peak energy of 19 keV; sample-to-detector distance of 10 cm and pixel size of 0.9 µm. The total number of projections for each sample was 2048, while the analyzed field of view (FOV) was 2048 × 2048 px^2^, equivalent to an acquired volume of about 6.3 mm^3^, a relatively small portion with respect to the entire scaffold volume (V_nom_ = 500 mm^3^). Tomographic slice reconstruction was performed using SYRMEP Tomo Project (STP) open-source software (Version 1.6.3) [33], developed by beamline researchers for its users. Paganin’s phase retrieval algorithm [34] has been applied. Ten cubic volumes of interest (VOIs) with a voxel edge of 270 µm were selected for each sample by FIJI and analyzed with VG Studio MAX 1.2 software (Volume Graphics GmbH, Heidelberg, Germany). Representative threshold values were manually set for PCL and HA and structural analysis was carried out on three-dimensional images. Dragonfly software (Version 2022.1) (ORS, Montréal, QC, Canada) was used to visualize the samples and produce 3D models of the inner areas.

## 3. Results

### 3.1. Powder Characterization

The results of SEM observations of PCL/HA feedstock powders are shown in Figure 2. The PCL and HA particles are discernible by shape and contrast, due to image acquisition with backscattered electrons. The PCL powder is formed of irregular structures (dark contrast), while HA has a roundish shape (bright contrast). From the normal size distribution, most of the HA particles were observed in the range of 15–25 μm, while PCL was observed in the 65–85 μm range (Figure 3).

XRD patterns of feedstock powders (P5, P30 and P50), in the full angular range investigated, are reported in Figure 4. Peak position of PCL is indicated by full dots, while all remaining peaks are due to hydroxyapatite Ca_10_(PO_4_)_6_(OH)_2_, hexagonal, with lattice parameters a = 0.93944 nm and c = 0.68751 nm (ICDD 74–565). Figure 4 shows well-defined and narrow peaks of all phases, suggesting good crystallization of PCL and HA powders. Moreover, the sequential increment in the relative intensity of HA peaks in P5, P30 and P50 patterns agrees with the increase in the relative amount of HA in the mixed powder.

### 3.2. Scaffold Characterization

#### 3.2.1. SEM Analysis

The SEM acquisition of scaffold skin-up surfaces is reported in Figure 5. HA is visible as bright round-shaped particles dispersed in PCL, which shows a gray contrast. After production, powder particles can be distinguished on the scaffold surface due to incomplete melting on the scaffold surface (Figure 5). The morphology of PCL particles is irregular and edgy, with particle boundaries easily distinguishable. This effect is less pronounced for scaffolds with 30 wt.% HA (Figure 5c,d). This was especially the case for RD30, where the particles were fully melted on the strut surface of the scaffold (Figure 5d). Although they appear unevenly distributed on the scaffold struts (Figure 5), HA particles (inset in Figure 5f) act as a solid dispersion in the PCL/HA scaffold matrix.

Ca/P atomic ratio, obtained by EDS analysis performed on DO and RD scaffolds as well as on the feedstock powders, is reported in Table 2.

XRD patterns of the scaffold top surface are reported in Figure 6. Patterns of DO and RD geometries are shown for the different HA amounts, vertically shifted to ease comparison. XRD patterns of scaffolds are reported in Figure 6 in the reduced angular range 2θ = 15°–45° where the most intense peaks of both phases (PCL and HA) are positioned. PCL peaks are indicated by full dots, while all other peaks are due to HA.

A qualitative comparison of XRD patterns of DO (Figure 6a) and RD (Figure 6b) scaffolds in terms of peak intensity and width suggests a better crystallization of the DO geometry.

#### 3.2.2. Haralick’s Surface Analysis

The results of the Haralick’s surface texture analysis are reported in Figure 7. From a descriptive statistics viewpoint (Figure 7a,b), the higher the HA percentage, the closer the values of energy and variance. It is, however, possible to achieve a multivariate view [35] by fitting a repeated measures model (Figure 7c), where the Haralick’s features together generate their marginal means (it is possible to show multivariate statistics in a univariate manner). For each geometry, the surface textures of 30 and 50 wt.% HA are statistically equivalent (*p* > 0.05). In addition, the higher the HA percentage, the closer the values of marginal means, indicating that the surface textures of DO50 and RD50 are statistically equivalent (*p* > 0.05). The other comparisons (within each geometry or between geometries with equal percentages of HA) were also significant (with at least *p* < 0.05).

#### 3.2.3. Roughness

Parameters related to surface roughness, derived from surface maps, are listed in Table 3. Although both geometries have a predominance of valleys (Ssk < 0), RD scaffolds present roughness profiles with a peak distribution (Sku > 3) that is higher and sharper with respect to values obtained in DO.

#### 3.2.4. XµCT

The scaffold morphometric parameters resulting from XµCT analysis are listed in Table 4. Values of micro-porosity and total porosity are considered for DO and RD scaffolds for each compound composition. Total porosity is higher in RD than in DO, except in the case of DO30 and RD30 where both geometries exhibited similar values. Micro-porosity within the scaffold struts, conversely, is higher in DO with respect to RD and in scaffolds with 30 and 50 wt.% HA, compared to 5 wt.%.

### 3.3. Mechanical Test

The results of mechanical tests performed on DO and RD geometry for each compound composition are plotted in Figure 8. Scaffolds with 30 and 50 wt.% HA exhibit similarly shaped load/compression curves. The curves show an initial steep rise, due to elastic compression of the struts in the unit cell, of up to about 10% compression with respect to the initial scaffold height. After elastic deformation, the curve shows a slowly increasing linear behavior, because of the progressive collapse of the struts (plateau). This plastic collapse is followed by a significant increase in the slope of the curve, coinciding with the densification of the scaffold. Compression tests were considered complete at about 50% of the specimen height. Scaffolds with 30 wt.% HA have a higher ultimate compressive strength than scaffolds with 50 wt.% HA. Scaffolds containing 5% wt.% HA exhibit a less evident plastic collapse regime, with the compaction of the unit cells starting at about 25% compression.

Values of scaffold elastic modul©(E) and ultimate compressive strength (σ_UC_), derived from compression curves in Figure 8, are reported in Table 5 for DO and RD geometries and 5, 30 and 50 wt.% HA.

It is worth noting that in the elastic regime the unit cell geometry has a predominant influence on the mechanical response of scaffolds only in samples with low HA amounts, while at concentrations of 30 and 50 wt.% the HA amount becomes the determining factor. In the plastic regime the behavior is dependent on both geometry and HA concentration. Densification is fully dependent on the material. Furthermore, at 50 wt.% HA the mechanical behavior of scaffolds is always determined by the material, regardless of unit cell geometry.

### 3.4. Cell Adhesion and Proliferation

Viability tests were carried out on DO and RD scaffolds containing different amounts of HA after 24 h (1 d) and 4 days of hMSC culture. In Figure 9, the data are represented as the number of cells at both culture times. It can be seen that the increasing addition of HA contributed positively to cell adhesion and proliferation of DO and RD geometries. Within the DO geometry, observed cell numbers were significantly higher at 30 and 50 wt.% HA, when compared to the 5% concentration (** *p* < 0.01), after 24 h incubation. Furthermore, an increase in cell proliferation was observed at 4 days compared to 24 h, although this was only significant in the sample with 5 wt.% HA (** *p* < 0.01) (Figure 9a). A similar trend was observed in the RD geometry, where cell numbers were significantly higher at 30 and 50 wt.% HA, compared with 5 wt.% HA, at both 24 h and 4 days of incubation (Figure 9b). A significant increment in cell growth was evident in all three concentrations at 4 days, compared to 24 h culture (Figure 9b). In terms of geometries, a significantly higher number of cells was determined on the RD geometry, at both 24 h and 4 days at all HA concentrations. This indicates the critical contribution of unit cell geometry in the interaction with cells.

In an effort to clearly understand the morphology and the behavior of the hMSCs, SEM observations (Figure 10) were carried out after 4 days of incubation on RD30 (Figure 10a) and RD50 and not on RD5 (Figure 10b), because of the greater number of cells in the former than in the latter. As depicted in Figure 8, hMSCs spread extensively on the inner struts of both scaffold typologies, extending their filopodia and cellular protrusions across the scaffold pores. It is interesting to note that SEM analysis after 4 days of culture revealed the presence of few hMSCs on scaffold surfaces, most likely as a result of cell colonization of the innermost scaffold layers.

### 3.5. PhC-XµCT

Hydroxyapatite weight percentages (wt.%), obtained by PhC-XµCT analysis, are reported in Table 6 for both scaffold geometries with 5, 30 and 50 wt.% HA, after 4 days of cell culture. It is worth noting that in both geometries, at high HA percentages (30 and 50 wt.%), it is possible to identify areas filled with deposits of worn away composite material, whereas at lower HA percentages the struts maintained their configurations and the porous portions remained hollow (Figure 11).

## 4. Discussion

A scaffold serving as a cell transplant device is an emerging approach to bone tissue engineering for regenerating damaged tissue. Previous works in the literature on PCL/HA biodegradable scaffolds produced by additive manufacturing have been focused on the effects of geometry or ceramic amount on scaffold biomechanical behavior, separately. This experimental work aims to analyze the combined effects of elementary unit cell geometry and HA dispersion on additively manufactured PCL/HA scaffold performances. PCL/HA scaffolds were produced by laser powder bed fusion (LPBF) technology with diamond (DO) and rhombic dodecahedron (RD) elementary unit cells (Figure 1) at concentrations of 5, 30 and 50 wt.% HA, from the perspective of bone tissue regeneration.

The manufacturing of scaffolds started by using the mix of PCL and HA powders illustrated in Figure 2, where the increasing content of round-shaped HA particles dispersed in the feedstock powders is clearly visible with the variation in the HA amount. Both phases (PCL and HA) in the mixed powder are well crystallized (Figure 4), with the Ca/P ratio of HA close to the expected stoichiometric value of 1.67 (Table 2), within uncertainties, as experimentally found by EDS analysis. The wide range of PCL particle sizes (blue curve in Figure 3), together with the large variability in density observed in [20], is related to the fairly large temperature range of PCL powder [20]. This variation results in a melting process which is difficult to control during the scaffold production by LPBF. The thermal behavior of PCL powder, along with the heat dissipation rate which is influenced by the elementary unit cell geometry, provokes a non-homogeneous melting of PCL particles, which are larger and denser, on the scaffold surface (Figure 5), as also found in [17,20].

Surface roughness (Sku) of DO and RD scaffolds with 5 wt.% HA shows similar values regardless of the geometry of unit cells (Table 3); while at higher HA concentrations (30 and 50 wt.%) the Sku parameter is influenced by both geometry and HA concentration (Table 3). The estimated marginal means of the Haralick’s texture features (Figure 7c) have the same variation pattern in the DO and RD scaffolds. This result indicates that both elementary unit cell geometry and HA concentration impact on scaffold surface texture, in agreement with roughness measurements (Table 3).

Scaffolds exhibit a double porous structure, formed of the designed macro-pores of elementary unit cell geometry and the micro-pores within the struts. Elementary unit cell geometry determines the arrangement of material struts (scaffold connectivity) and, consequently, the distribution of a macro-porous network (scaffold tortuosity), as already widely described in Gatto et al. [20]. According to [20], the higher tortuosity of DO geometry leads to a slower heat dissipation rate during scaffold fabrication. The heat dissipation rate affects the material crystallinity during solidification and cooling. As a consequence, the DO scaffold tends to have higher crystallinity of both PCL and HA with respect to RD, regardless of the HA amount (Figure 6). As already demonstrated in our previous work [20], scaffold crystallinity depends on unit cell topology, while scaffold chemical composition is independent of geometry (Table 2). Micro-porosity within the struts is controlled by HA concentration and elementary unit cell geometry (Table 4). The latter affects the global heat dissipation rate of the scaffold, while the local mechanism of heat dissipation depends on the amount of HA particles that remain unfused during the manufacturing process.

Mechanical compression tests revealed a different elastic behavior depending on the HA concentration (Figure 8). With only 5 wt.% HA, the elastic regime was influenced by the unit cell geometry (black curves in Figure 6 and quantification in Table 5). On the other hand, for higher HA concentrations (30 and 50 wt.%), the elastic deformation regime was fully controlled by the HA amount, since DO and RD geometries show similar behavior in terms of load/compression curve shape (Figure 8) and elastic modulus values (Table 5). In more detail, the contribution of unit cell geometry in the elastic regime is associated with the elastic response of struts forming the cells of the lattice structure. Moreover, elastic modulus can be increased by adding HA particles, as found by Rezania et al. [21] in extruded scaffolds in PCL/HA (from 5 to 20 wt.%). However, this trend shows a threshold between 30 and 50 wt.% HA concentrations (Table 5). This result extends to the elastic regime previous findings of Eosoly et al. [17], showing that the addition of HA into the powder mix caused a decline in mechanical properties (compressive strength) of LPBF scaffolds with PCL/HA up to 30 wt.%. The effect of HA incorporation on the mechanical behavior of scaffolds depends on the sintering quality of the polymer powder particles. In the case of weak polymer sintering, HA particles could not properly integrate into the polymeric matrix, acting as an insulating agent working against coherent sintering within layers and against cohesion between layers [17]. According to the findings of Gatto et al. [36] on polyamide (PA) filled with aluminum or alumina particles, the reinforcing mechanism is effective in two circumstances: (a) nano-reinforcements and (b) strong bonds between matrix and filler. In all other cases, the filler is ineffective or even detrimental to the mechanical performances.

Plastic deformation (plateau in Figure 8) is controlled by both HA concentration and elementary unit cell geometry. Since DO geometry presents a more connected and open-pore structure than RD [20], it requires a higher load to achieve the same plastic deformation values, especially with up to 30 wt.% HA. At 50 wt.% HA, conversely, it is the material composition which mostly influences plastic collapse, regardless of geometry.

During scaffold densification, when pores are flattened and struts broken, the material properties determine the mechanical response of the scaffold [20], leading to a decrease in the ultimate compressive strength (σ_UC_) as HA amounts are increased (Table 5). Therefore, higher amounts of HA in the PCL/HA composite scaffolds induce deterioration of the mechanical properties of the scaffold, in good agreement with the mechanical results of Eosoly et al. [17].

However, regardless of unit cell geometry (DO or RD) or variations in HA concentration (5, 30 or 50 wt.%), all scaffolds presented an ultimate compressive strength which was in the range of human mandibular trabecular bone [37].

The biological performances of scaffolds were tested with short-term hMSC culture (Figure 9). Cell adhesion after 24 h is significantly higher in 30 and 50 wt.% HA than 5 wt.%, regardless of unit cell geometry (Figure 9a,b). This suggests that, since scaffolds exhibit a similar surface pattern (Figure 5), hMSC adhesion after 24 h of incubation is affected more by material composition than by surface texture. On the contrary, Liu et al. [22] did not observe any significant differences in cell proliferation activity among all scaffolds (0–25 wt.% HA in PCL matrix) after 1 day, whereas evident differences in cell proliferation activity were observed after 3 days of culturing, with 25 wt.% PCL/HA possessing the highest proliferation of MC3T3-E1 osteoblasts [22]. After 4 days of hMSC culturing, RD30 and RD50 resulted as the most favorable environments for cell proliferation (Figure 9b). This is likely due to the more connected and open-pore structure, which allows cell colonization across the whole structure because of the appropriate supply of nutrients and oxygen [20]. After 4 days culturing, cell viability is largely determined by the elementary unit cell geometry. The absence of hMSCs on the top surface of scaffolds after 4 days of incubation (Figure 10) suggests cell migration from the surface to scaffold inner layers.

To further understand the interaction between cells and material, phase contrast X-ray micro-tomography (PhC-XµCT) by synchrotron radiation analysis was performed on scaffolds after 4 days of culture (Table 6 and Figure 11). The PCL degradation, in addition to culture medium and presence of cells, is due to the combined effects of elementary unit cell geometry, micro-porosity, PCL molecular weight and the amount and size of HA, as extensively demonstrated by Gatto et al. [20]. The experimental quantification of average HA amounts in PCL scaffolds with DO and RD geometries shows high standard deviation (SD) values (Table 6). These high SD values are probably imputable to localized degradation phenomena, in turn attributable to (a) an inhomogeneous distribution of HA (as suggested by Eosoly et al. [17] and observable in Figure 4) and (b) a poor control of micro-pore dispersion within the struts, as shown by 2D cross-sectional slices in Figure 11a,b. As a result, it cannot be quantitatively demonstrated that the HA concentration generally increases because of the PCL degradation. However, 3D reconstruction of RD5 in Figure 11c clearly displays open macro-pores, while RD30 macro-pores are filled with eroded fragments composed mainly of PCL (Figure 11d). This result aligns with the RD5 and RD30 micro-porosity values in Table 4. The higher micro-porosity of RD30 (Figure 11b) enables the biomaterial degradative process to accelerate, allowing for fluid infiltration within the scaffold struts [20].

## 5. Conclusions

The aim of this work was to fill the gap in the literature of the combined effects of elementary unit cell geometry and HA amounts on the performances of additively manufactured PCL/HA scaffolds for tissue engineering applications. Scaffolds were produced by laser powder bed fusion (LPBF) with diamond (DO) and rhombic dodecahedron (RD) elementary unit cells, and 5, 30 and 50 wt.% HA.

Scaffold biomechanical and degradative behaviors were investigated by structural, mechanical and biological characterizations, to evaluate the influence of elementary unit cell geometry and HA concentration on the scaffold performances. The main results can be summarized as follows:Mechanical tests evidenced three deformation regimes: elastic, plastic and densification:
○In the elastic regime, geometry of the elementary unit cell governs the behavior of the scaffold containing the lowest amount of HA (5 wt.%), while at higher concentrations (30 wt.% and 50 wt.% HA) the mechanical response depends on the material properties;○In the plastic regime, for scaffolds with 5 and 30 wt.% HA, the DO geometry requires higher load to achieve the same deformation value of RD, while for scaffolds with 50 wt.% HA the load values are almost the same;○In the densification regime, the ultimate compressive strength decreases with an increasing of HA amount;
Regardless of elementary unit cell geometry (DO or RD) or HA amount (5, 30 or 50 wt.%), scaffolds presented an ultimate compressive strength value in the range of human mandibular trabecular bone;The biological response of hMSCs after 24 h of culture is mostly affected by material composition, with enhancement of cell adhesion for HA amounts of 30 and 50 wt.%;After 4 days of culture, cell viability is affected by elementary unit cell geometry, with RD30 and RD50 being the most favorable environments for cell proliferation. Moreover, higher micro-porosity values accelerate the biomaterial degradative process.

## Figures and Tables

**Figure 1 materials-16-04950-f001:**
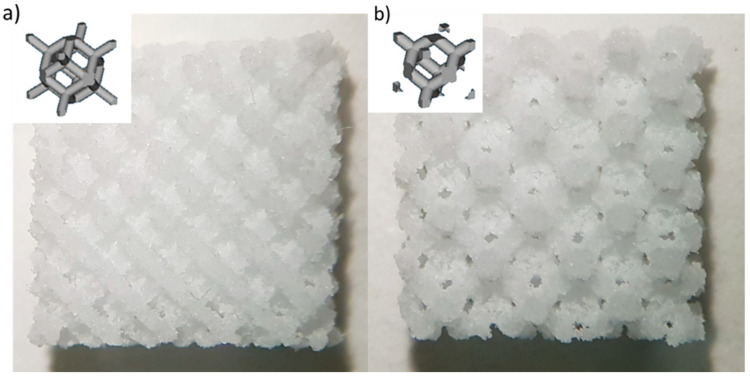
Images of the produced scaffolds: (**a**) Diamond (DO) geometry and (**b**) rhombic dodecahedron (RD) geometry. Schematic of the elementary unit cell geometry is shown in the inset for each scaffold type.

**Figure 2 materials-16-04950-f002:**
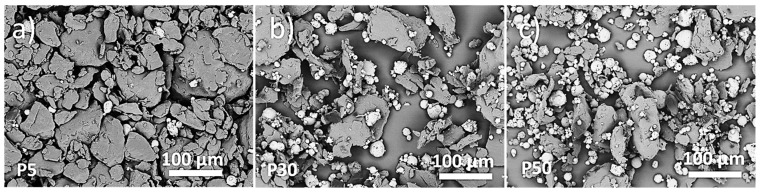
SEM micrographs of PCL/HA feedstock powders: P5 (**a**), P30 (**b**) and P50 (**c**).

**Figure 3 materials-16-04950-f003:**
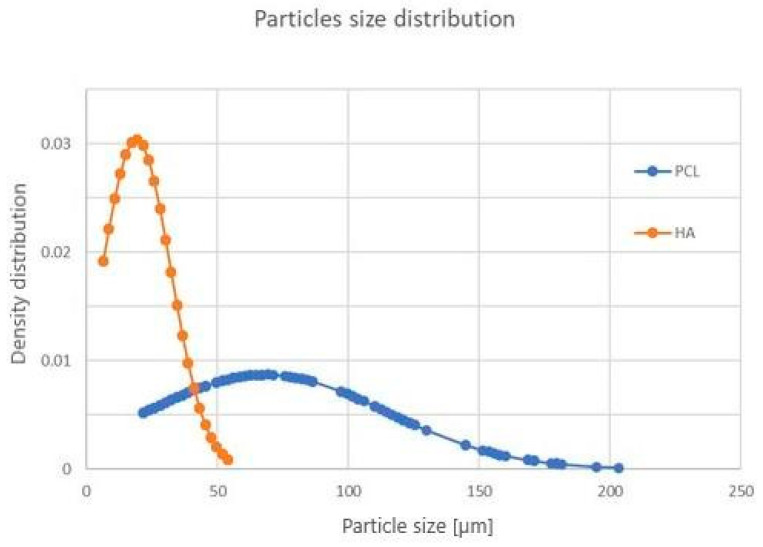
Particle size distribution of PCL (blue curve) and HA (orange curve) powder.

**Figure 4 materials-16-04950-f004:**
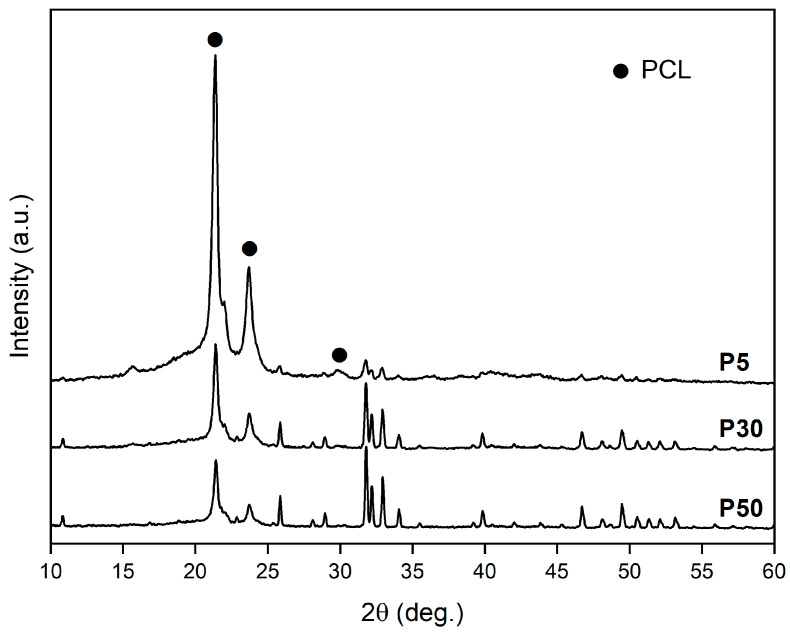
XRD patterns of P5, P30 and P50 powders. PCL—full dot. All other peaks are due to HA.

**Figure 5 materials-16-04950-f005:**
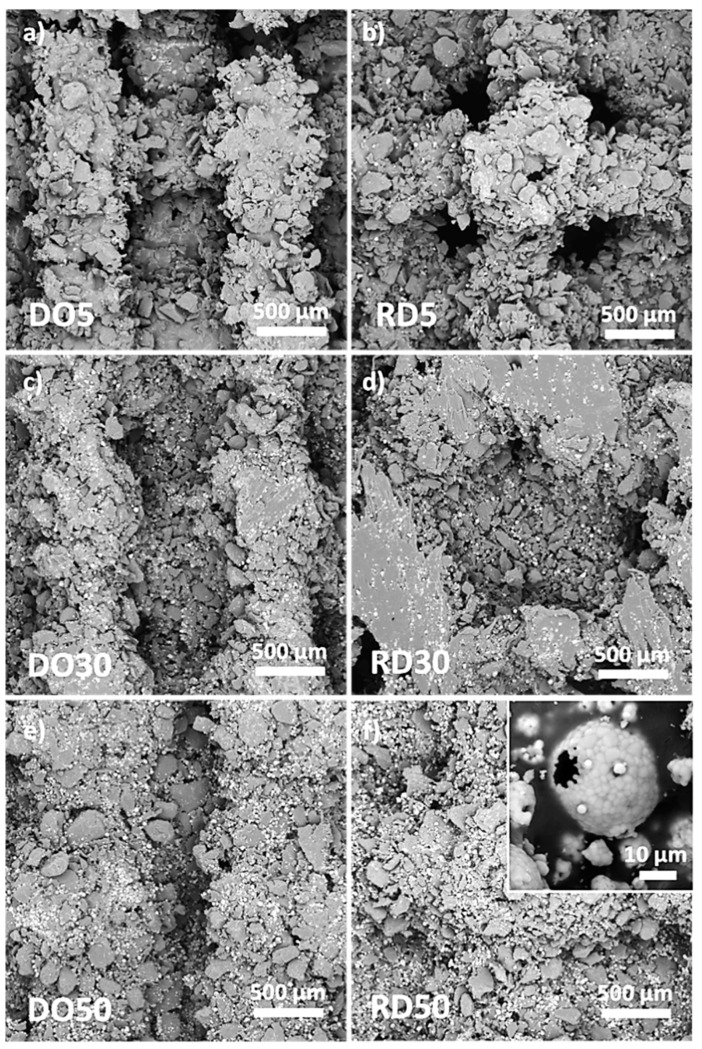
SEM images of surfaces of DO and RD PCL/HA scaffolds with 5 wt.% (**a**,**b**), 30 wt.% (**c**,**d**) and 50 wt.% (**e**,**f**) HA. Inset in (**f**) shows details of a hydroxyapatite particle.

**Figure 6 materials-16-04950-f006:**
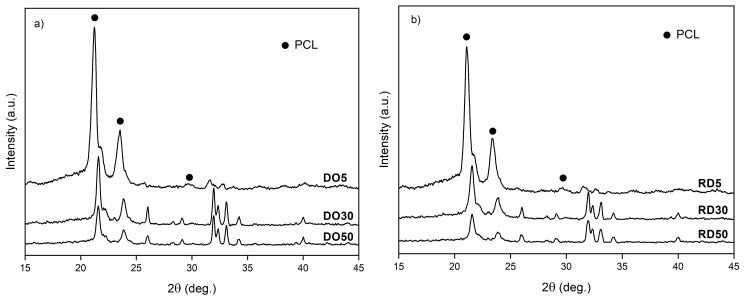
XRD patterns of DO (**a**) and RD (**b**) scaffolds with 5 wt.%, 30 wt.% and 50 wt.% HA. PCL—full dots, HA—all other peaks.

**Figure 7 materials-16-04950-f007:**
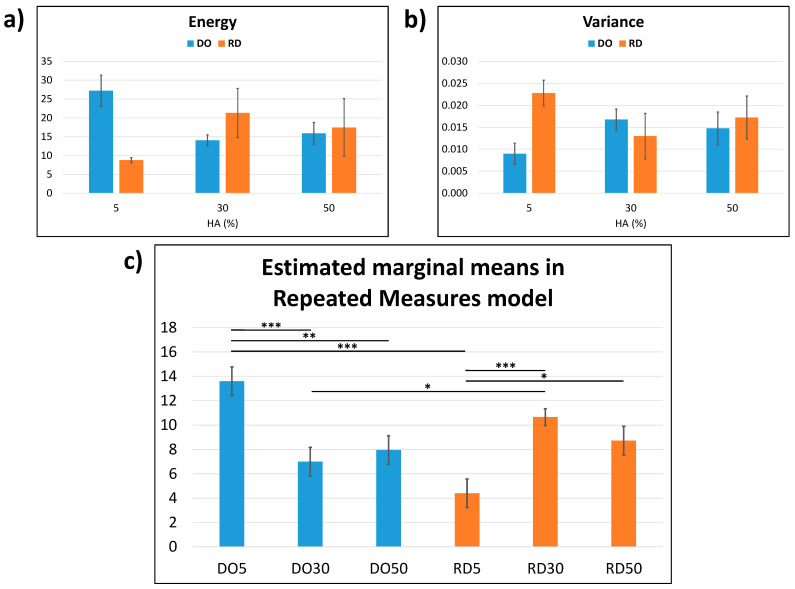
Haralick’s surface texture analysis: (**a**) Haralick’s energy; (**b**) Haralick’s variance and (**c**) estimated marginal means in repeated measures model calculated from (**a**,**b**). The results are presented as mean ± SD in (**a**,**b**), whereas as mean ± standard error in (**c**) with * *p* < 0.05, ** *p* < 0.01, *** *p* < 0.001 (the comparisons were made within each geometry or between geometries with equal percentages of HA).

**Figure 8 materials-16-04950-f008:**
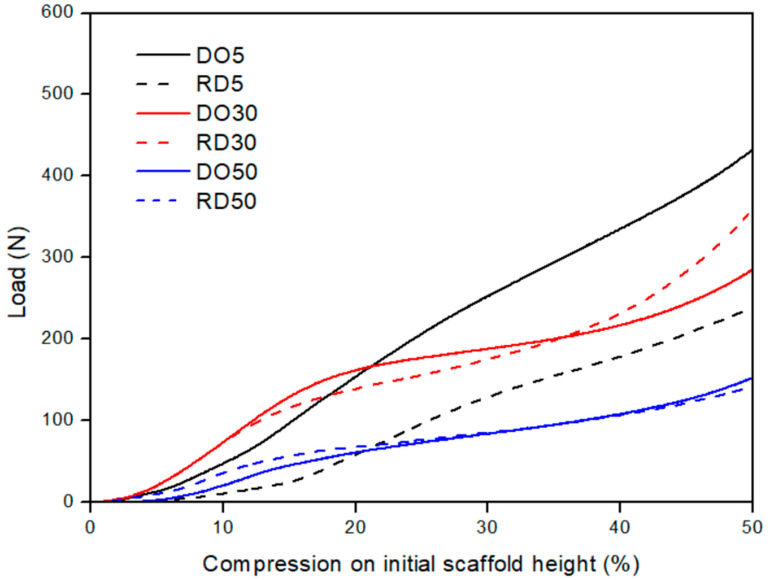
Mechanical performances of DO and RD PCL/HA scaffolds with 5 (black curves), 30 (red curves) and 50 (blue curves) wt.% HA. DO—straight line; RD—dashed line.

**Figure 9 materials-16-04950-f009:**
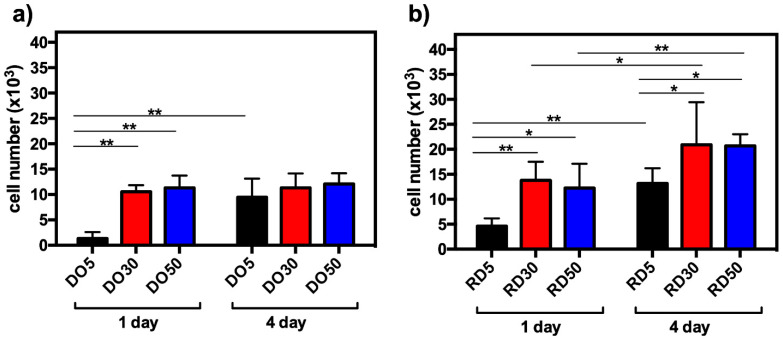
hMSC viability after 24 h (1 d) and 4 days (4 d) of culture on DO (**a**) and RD (**b**) scaffolds with 5, 30 and 50 wt.% HA, * *p* < 0.05 and ** *p* < 0.01.

**Figure 10 materials-16-04950-f010:**
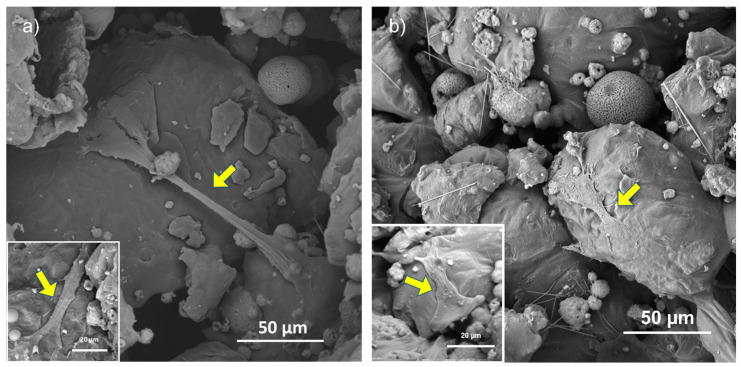
SEM observation of scaffold inner surface of RD30 (**a**) and RD50 (**b**) scaffolds. Yellow arrows point to cell in adhesion on scaffold struts.

**Figure 11 materials-16-04950-f011:**
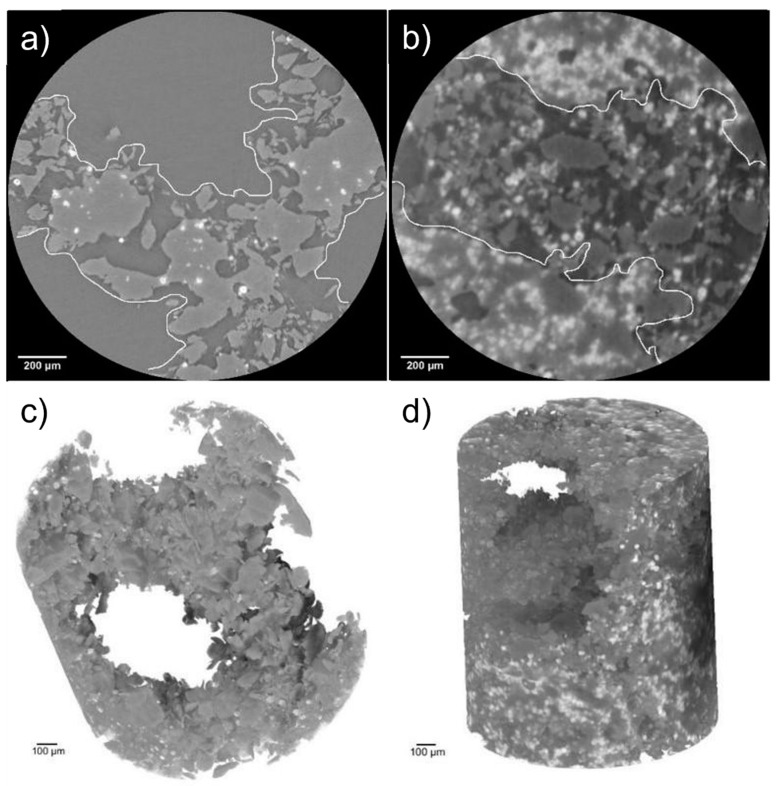
RD5 (**a**) and RD30 (as example for high HA percentage) (**b**) with different distributions of struts and empty portions; 3D models of RD5 (**c**) and RD30 (**d**) acquired at high resolution.

**Table 1 materials-16-04950-t001:** Experimental settings for XµCT analysis.

Phase	Parameter	DO5	RD5	DO30	RD30	DO50	RD50
Acquisition	Pixel size [μm]	11.5					
	Rotation step [deg.]	0.4 for 180					
	Frame averaging	2					
	Al filter [mm]	No filter		0.25		0.25	
	Exposure time [s]	1.6		3		4	
Reconstruction	Smoothing	1		2		2	
	Ring artifact reduction	2		2		2	
	Beam hardening correction [%]	5		20		25	

**Table 2 materials-16-04950-t002:** Results of the EDS chemical analysis for Ca/P atomic ratio of powders, DO and RD scaffolds with 5, 30 and 50 wt.% of HA.

		Ca/P (at.%)	
	HA 5%	HA 30%	HA 50%
Powder	1.6 ± 0.2	2.1 ± 0.1	2.1 ± 0.1
DO	1.9 ± 0.1	2.3 ± 0.4	2.1 ± 0.2
RD	1.9 ± 0.1	1.7 ± 0.3	2.1 ± 0.1

**Table 3 materials-16-04950-t003:** Roughness parameters from roughness maps.

Parameter		DO			RD		
		DO5	DO30	DO50	RD5	RD30	RD50
Ssk	Symmetry of roughness profile with respect to mean line	−0.5	−0.24	−1.3	−0.6	−0.4	0.2
Sku	Sharpness of roughness profile	2.6	2.4	4.4	2.8	4.28	3.2

**Table 4 materials-16-04950-t004:** Scaffold morphometric parameters obtained from XµCT analysis.

Parameter	DO			RD		
	DO5	DO30	DO50	RD5	RD30	RD50
Total porosity [%]	65 ± 1	56 ± 4	64 ± 2	70 ± 1	53 ± 1	69.5 ± 0.5
Scaffold micro-porosity [%]	1.9 ± 0.1	5.9 ± 0.4	4.6 ± 0.4	1.4 ± 0.1	3.7 ± 0.3	3.1 ± 0.1

**Table 5 materials-16-04950-t005:** Mechanical parameters from uniaxial compressive test. E—elastic modulus; σ_UC_—ultimate compressive strength.

Mechanical Parameter	DO			RD		
	DO5	DO30	DO50	RD5	RD30	RD50
E (MPa)	10.8 ± 0.6	13 ± 2	8 ± 2	7.2 ± 0.5	11.6 ± 0.7	6.3 ± 0.1
σ_UC_ (MPa)	2.8 ± 0.7	2.20 ± 0.05	1.5 ± 0.6	1.9 ± 0.1	2.4 ± 0.2	1.07 ± 0.01

**Table 6 materials-16-04950-t006:** HA wt.% average value and standard deviation in PCL/HA scaffolds, after 4 days of cell culture.

	DO			RD		
	DO5	DO30	DO50	RD5	RD30	RD50
HA [wt.%]	9 ± 2	50 ± 20	70 ± 20	8 ± 1	50 ±10	70 ± 20

## Data Availability

The data presented in this study are available on request from the corresponding author.

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
