# Peer review of "Combined Effects of HA Concentration and Unit Cell Geometry on the Biomechanical Behavior of PCL/HA Scaffold for Tissue Engineering Applications Produced by LPBF"

_materials, 2023, doi:10.3390/ma16144950_

Round 1

Reviewer 1 Report

Manuscript entitled “Combined effects of HA concentration and unit cell geometry on the biomechanical behavior of PCL/HA scaffold for tissue engineering applications produced by LPBF”, by Michele Furlani, et al., investigated the effect of geometry and HA amount on the biomechanical and degradative behaviour of PCL/HA scaffolds. Morphological and structural of PCL/HA powders, the scaffold surface and cells were investigated by SEM et. al. Results clearly showed the impact of HA amount and unit cell geometry on the mechanical response as well as on the biological and degradative behavior. The results showed in the manuscript are interesting, but there are still some problems:

1. In the abstract and conclusion sections, the author needs further condense and summarize.

2. Overall discussion is ok however it should be improved, especially SEM analysis of PCL/HA feedstock powders and Scaffold characterization.

3. In 2.2. morphological and structural characterization, the authors mentioned EDS and XRD tests, however they were not described in results. As a result, EDS and XRD tests of PCL/HA feedstock powders and Scaffold should be added.

4. There are some errors in the manuscript (e. g. P. 8, Line 251: “Figures 4”; P. 10, Line 312: “Mpa”, …….), the author should carefully check and make them correct.

5. The format for the references should be standardized.

Author Response

Dear reviewer,

We would like to thank you for your suggestions that will help us to improve the quality of the article.

Please see the attachment below.

Regards.

Reviewer 2 Report

After read carefully this present manuscript, I would like to recommand this article for publication in your journal Materials after some modifications:

The authors studied the appareance of the PCL/HA powder at different ratios. In fact, it should be interesting to observe the behavior of P50 when heated  at the higher temperatures. Then, simple XRD analysis they could also mesure the impact of HA addition on the global morphology of the material at room temperature. Maybe try to analyze the material by FT-IR could be interesting (or if it was ready dexcribed eleswhere in the literature).

It should be interesting to add a picture of the entire scaffold.

To promote the proliferation and groth of a target cell agood material needs porosity and connectivity. Is it possible to add more informations about these aspects.

Enough Rigidity of the material is synonym of bone cell differentiation,, however the hydrophobic:hydrophilic balance of the material could be also connected to cell adherance. Did the authors try to determine these characteristics with contact angle messurement ?

I think this is correct.

Author Response

(The authors gave the same response as above.)

Round 2

Reviewer 1 Report

The revised manuscript has addressed the issues proposed by the reviewer. Now it can be accepted in this current version.